# Asymptomatic Type 2 Perioperative Myocardial Infarction Detected before Anesthetic Induction in a Patient Undergoing Non-Cardiac Surgery—A Case Report

**DOI:** 10.3390/medicina59061168

**Published:** 2023-06-17

**Authors:** Seunghee Ki, Soo Jee Lee, Juseok Oh, Seung Bae Cho, Seongmin Park, Taesoo An, Jeonghan Lee

**Affiliations:** Department of Anesthesiology and Pain Medicine, Busan Paik Hospital, Inje University College of Medicine, Busan 47392, Republic of Korea; 108873@paik.ac.kr (S.K.); 109414@paik.ac.kr (S.J.L.); 093724@paik.ac.kr (J.O.); 089517@paik.ac.kr (S.B.C.); 093715@paik.ac.kr (S.P.); 094199@paik.ac.kr (T.A.)

**Keywords:** brain natriuretic peptide, coronary angiography, coronary artery disease, intraoperative monitoring, myocardial infarction, perioperative period, troponin

## Abstract

*Background*: Perioperative myocardial infarction (PMI) is a life-threatening complication in major non-cardiac surgeries (NCS) and constitutes the most common cause of postoperative morbidity and mortality. A PMI that is associated with prolonged oxygen supply–demand imbalance and its etiology is defined as a type 2 MI. Asymptomatic myocardial ischemia can occur in patients with stable coronary artery disease (CAD), especially those with comorbidities such as diabetes mellitus (DM), hypertension, or, in some cases, without any risk factors. *Case*: We report a case of asymptomatic PMI in a 76-year-old patient with underlying hypertension and DM without a previous history of CAD. During the induction of anesthesia, abnormal electrocardiography was discovered, and the surgery was postponed after further studies revealed almost completely occluded three-vessel CAD and type 2 PMI. *Conclusions*: Anesthesiologists should closely monitor and evaluate the associated cardiovascular risk, including cardiac biomarkers of each patient before surgery, to minimize the possibility of PMI.

## 1. Introduction

Perioperative myocardial infarction (PMI) is a leading cause of mortality and morbidity in the elderly undergoing non-cardiac surgeries (NCS). The elderly with comorbidities such as diabetes mellitus (DM) and hypertension have a higher risk of PMI. However, a diagnosis is often missed because patients with stable coronary artery disease (CAD) have shown more often to be asymptomatic, rather than symptomatic, for ischemia [1]. Furthermore, angina pectoris itself is now considered to poorly indicate myocardial ischemia, especially in patients with DM [2]. PMI can develop at any moment of the perioperative period, and asymptomatic progression can remain undetected by cardiologists, especially after the preoperative evaluation has been completed but before the surgery. For this reason, anesthesiologists must be able to evaluate the cardiovascular risk for each patient through a comprehensive evaluation of medical history, physical examination, laboratory values, and radiological examinations, and remain vigilant for the possible development of PMI. In this case report, we present a patient with hypertension and DM who was cleared for surgery by a cardiologist and endocrinologist as part of the patient’s preoperative assessment. We suspected asymptomatic myocardial ischemia in the operating room just before the administration of anesthetic drugs and the surgery was postponed. Further cardiac evaluations revealed almost completely occluded three-vessel CAD that required a coronary artery bypass graft (CABG).

## 2. The Case Description

A male patient, aged 76 years, presented with low back and left lower limb pain to the neurosurgery spine center of our hospital. The patient was diagnosed with spinal and foraminal stenosis on his L4/5 and was scheduled for oblique lateral lumbar body fusion (OLIF) surgery. His height and weight were 166.2 cm and 61.9 kg, respectively. The patient had a medical history of hypertension and insulin-dependent DM and has been taking medication for over 30 years with regular biannual visits to the endocrinologist and cardiologist. A calcium channel blocker (CCB), angiotensin II receptor blocker, and hydrochlorothiazide were prescribed by the cardiologist for hypertension. Metformin and combined basal and rapid insulin were prescribed by the endocrinologist for insulin-dependent DM. His surgical history includes a distal pancreatectomy 40 years ago due to pancreatitis.

The patient underwent routine preoperative examinations 15 days before the surgery, including electrocardiography (ECG), trans-thoracic echocardiography (TTE), chest X-ray, and laboratory blood tests. The resting 12-lead ECG recorded normal sinus rhythms with left ventricular hypertrophic patterns (Figure 1). The preoperative TTE was unremarkable without any valvular abnormalities or restricted wall movement. The left ventricle ejection fraction (LVEF) was measured to be 57%. His chest X-ray showed fibrotic scars in the bilateral upper lobes, suggestive of inactive tuberculosis lesions. The laboratory results were within normal limits, except for a slightly elevated HbA1c level of 6.8%. The patient underwent consultations with a cardiologist, a pulmonologist, and an endocrinologist, and, due to the high risk of systemic diseases, was classified as the American Society of Anesthesiologists Physical Status Classification III. Prior to entering the operating room, the patient was premedicated with intramuscular injections of glycopyrrolate (0.2 mg) and famotidine (20 mg).

Upon entering the operating room, the patient was monitored with augmented limb ECG with continuous ST-segment analysis, non-invasive blood pressure (NIBP) (MX700, Philips Healthcare), pulse oximetry (Nellcor™ SpO_2_ Adhesive Sensors, Medtronic), and bispectral index (BIS™ Quatro, Medtronic). Initial vital signs prior to induction of general anesthesia were: NIBP 142/60 mmHg, heart rate 81 beats/min, and SpO_2_ 95%. ST-segment depression and T-wave inversion were observed on ECG leads II and III. The ST-segment numeric value analysis at the J 60 point of leads II and III were −2.3 and −2.2, respectively (Figure 2). At this point, the induction phase of anesthesia was put on hold, and venous blood samples were drawn from the patient for cardiac biomarkers. An additional precordial V5 lead was placed, and a 12-lead ECG was performed for immediate evaluation. ST-segment depressions were also observed on leads V4, V5, and V6, while the remaining leads were ambiguous. 

The patient was not sedated and was closely observed in the surgery room until further evaluation could be made with a cardiologist. Vital signs, including the ECG, were continuously monitored. The patient denied any symptoms of angina or anginal equivalents, such as dyspnea, light-headedness, fatigue, or gastrointestinal symptoms. During this period, we administered sublingual nitroglycerine (NTG) at 0.6 mg, administered a continuous intravenous infusion of remifentanil at 0.04 mcg/kg/min, and supplied oxygen via a nasal cannula with the flow of 3 L/min. Beta-blockers or CCBs were not administered. Ten minutes after the administration of NTG, the ST segments of leads II, III, and V5 improved (Figure 3). The surgery was postponed, and the patient was transferred to the cardiology center for further evaluation.

Emergency TTE, coronary angiography (CAG), and computed tomography angiography (CTA) were performed at the cardiology center. The TTE revealed that the patient’s ventricular function was preserved with an LVEF of 68% and did not show any valvular abnormalities. However, the CAG report indicated that the patient had three-vessel CAD with multiple diffuse stenoses. The right coronary artery, left circumflex artery, and left anterior descending artery were found to have stenoses of up to 90%, 100%, and 80%, respectively (Figure 4). The percutaneous coronary intervention was not possible due to the complexity of the case and the presence of three-vessel CAD with stenoses above 70%. The CTA also revealed diffuse atherosclerosis, with multiple atheromas along the thoracoabdominal aorta, and severe (>70%) intramammary artery stenosis. Cardiac biomarker levels that were sampled in the operating room later confirmed elevated cardiac troponin I (cTnI) and lactate dehydrogenase (LDH) with levels measuring 21.5 pg/mL and 225 U/L, respectively. Creatinine kinase-MB (CK-MB) was at 2.82 ng/mL, which was within the normal limit. The patient was diagnosed with type 2 MI and later transferred to the cardiothoracic surgery department to undergo elective CABG surgery a week later. The surgery was uneventful, and the patient was discharged 3 weeks later after recovery.

Following discharge from the hospital, the patient regularly visited the cardiothoracic surgeon and the endocrinologist for monthly follow-up examinations. During his last two visits, there were no ST changes in 12-lead ECGs, and his cardiac biomarkers were within normal ranges, including CK-MB, cTnI, and LDH. Brain natriuretic peptide (BNP) and N-terminal pro-brain natriuretic peptide (NT-proBNP) were not measured. The patient is well-ambulated, and his back pain is being managed at our pain clinic center through medication and nerve root block injections. The cardiothoracic surgeon will evaluate the patient’s conditions 12 months after his CABG for elective spine surgery with the neurosurgery department.

## 3. Discussions

PMI is considered an important complication in major NCS and is associated with poor prognosis. Patient-related risk factors for PMI in major NCS include the male sex, older age, the functional capacity of the patient, and the presence of cardiovascular risk factors, such as smoking, hypertension, diabetes, dyslipidemia, family disposition, or established cardiovascular disease. Furthermore, comorbidities such as heart failure, atrial fibrillation, and chronic renal failure, along with abnormal laboratory findings, have been identified as contributing factors [3,4]. However, most patients who experience PMI do not report ischemic symptoms as a result of anesthesia sedation or pain-relieving medications [5]. According to Schang SJ et al., asymptomatic ST-depression during ambulatory ECG monitoring occurs more frequently than symptomatic ST-depression in patients with CAD [1]. Rueztler et al. report only 14% of patients who experience PMIs report chest pain, and 65% of these events are clinically completely asymptomatic [6]. Furthermore, asymptomatic myocardial ischemia is reported in patients with mild-to-moderate hypertension, non-insulin-dependent DM, and even without risk factors regardless of surgery [1]. 

Currently, the *Fourth Universal Definition of Myocardial Infarction* in 2018 classifies two major causes of PMI, termed type 1 and type 2 MIs [5]. Type 1 MI is associated with atherosclerotic plaque rupture, with or without thrombus occlusion, while type 2 MI is associated with an imbalance between myocardial oxygen supply and demand [5]. The etiologies of type 2 MI include non-ruptured atherosclerosis, vasospasm, coronary dissection, and oxygen imbalance alone [5]. Type 2 MI can be triggered by emotional stress, exercise, anemia, hypotension, or sustained tachyarrhythmia in patients with known or suspected stable CAD [5,7]. According to Ruetzler et al., PMIs associated with NCS that is apparently caused by an oxygen supply and demand imbalance are considered type 2 MIs [6]. We believe that the preoperative period may have contributed to moments of heightened anxiety for our patient, triggering an increase in myocardial oxygen demand, leading to an imbalance in myocardial oxygen supply and demand, and subsequent type 2 MI or non-ischemic myocardial injury. His blood sample in the operating room revealed a cTnI level above the 99th percentile of the upper reference limit in addition to the ECG showing significant ST changes in multiple leads. The patient was diagnosed with type 2 PMI with three-vessel CAD as well as the presence of atherosclerosis on CAG.

The prevalence of PMI in NCS ranges from 3% to 6% [6], making thorough preoperative evaluation essential in patients with cardiovascular risk factors. Ruetzler et al. demonstrated the cost-effectiveness of routine troponin monitoring in surgical inpatients who are 45–64 years old and have at least one cardiovascular risk factor, as well as in all surgical inpatients who are 65 years old or older [6]. The Canadian Cardiovascular Society Guideline recommends measuring daily troponin for 3 days in patients with moderate cardiovascular risk and preoperative BNP, and NT-proBNP in patients 45–64 years of age with significant cardiovascular disease, or in those who are 65 years old or older [6,8]. The European Society of Cardiology additionally recommends the preoperative measurement of BNP and NT-proBNP in patients with risk factors or symptoms for intermediate-to-high-risk NCS [3]. Moreover, the American College of Cardiology and the American Heart Association recommend postoperative troponin surveillance in high-risk surgical patients with signs or symptoms of PMI [6,9]. Our patient had not been previously tested for cardiac biomarker levels, despite the patient’s multiple cardiovascular risks, because his regular check-up visits with internists indicated that his hypertension and diabetes were under control, and his preoperative TTE results were also normal. However, through ECG monitoring in the operating room, we were able to suspect an asymptomatic PMI.

Anesthesiologists should also take caution when taking prophylactic measures to lower the risk of PMI. The Perioperative Ischemic Evaluation Study I (POISE-I) trial examined the impact of metoprolol on cardiovascular events and reported a 26% reduction in non-fatal PMI. However, the POISE-I trial also found an increase in mortality of 31% and in stroke of 100%, mainly due to hypotension and bleeding. Therefore, the consensus is to continue long-term beta-blocker usage, and newly starting prophylaxis for avoiding PMI is not recommended [6,7]. Additionally, POISE-2 and the Management of Myocardial Injury After Noncardiac Surgery (MANAGE) trial recommended against the de novo use of aspirin prophylaxis of PMI and resulted in the safer use of dabigatran compared with warfarin [6].

During the intraoperative period, it is essential to determine the appropriate treatment options for patients with acute PMI presentations. Nitrates, narcotic analgesics, beta-blockers, and CCBs are commonly recommended during the acute phase of MIs [10]. Many prophylactic measures that are commonly used in ambulatory settings such as a statin, aspirin, dual antiplatelet therapy (DAPT), and coronary revascularization can be applied intraoperatively; however, their application may not be as straightforward [11]. In patients with suspected PMI or those at risk of PMI, the primary goal for anesthesiologists is to treat all potential causes of tachycardia, hypertension, hypotension, and anemia, while also aiming to minimize postoperative pain [4,7,11]. In the case of type 2 MI, correcting the imbalance between myocardial oxygen supply and demand is crucial, and the early usage of beta-blockers to reduce myocardial demand should be considered [12]. If variant angina is suspected, CCBs are a valid option. Conversely, if there are no clear triggers for type 2 MI, if the presence of CAD is unknown, if there is the presence of ST elevation, or if there are changes in the cTn level, anesthesiologists must also consider the possibility of type 1 MI [12]. In our case, upon gaining the impression of asymptomatic PMI through the ECG monitoring in the surgery room, we promptly started administration of nitrates, opioids, and oxygen to maintain coronary perfusion. We also compared the ECG to the preoperative ECG that was taken 15 days ago for any missed signs of ST depression. However, despite the feasibility and availability of focused cardiac ultrasound, which could have aided our decision-making for MI, we were not able to proceed with TTE in the operating room due to a lack of skilled personnel.

Not all myocardial injuries require emergency interventional or surgical treatment. If there is a stable pattern of elevated cTn levels, it may indicate a chronic myocardial injury caused by underlying diseases such as heart failure or renal failure. This can be assessed through structural imaging techniques using echocardiography and laboratory tests, including renal function tests [12]. However, if dynamic patterns such as fluctuations in cTn levels are observed, further myocardial evaluations are necessary to avoid underdiagnosing possible acute MIs. Further evaluations may include taking a detailed medical history, conducting physical examinations, monitoring cardiac biomarkers through follow-up laboratory tests, and performing coronary angiography [12]. In cases where ST changes are observed in multiple ECG leads, culprit coronary lesions are detected in coronary angiography, unstable vital signs are present, and typical symptoms are reported, it may indicate a type 1 MI, which requires guideline-directed therapy for acute MI [3,4,12]. Treatment for type 1 MI involves coronary revascularization, DAPT, high-intensity statin therapy, and the administration of anti-hypertension medications [4]. In cases where there is compelling evidence of a type 2 MI, the focus shifts towards treating the underlying causes, such as coronary spasm, dissection, or tachyarrhythmias, and initiating therapy for CAD [12].

## 4. Conclusions

This case presents our encounter with an asymptomatic type 2 PMI patient who was not suspected of CAD until the patient monitoring process during the perioperative phase for anesthetic induction. Mortality and morbidity are significantly increased by PMI, underscoring the importance of early detection through patient symptoms, vital signs, ECG, imaging studies, and laboratory testing. However, detection of PMI is often difficult because it can manifest without any symptoms, especially in patients with DM. Furthermore, PMI can still occur after the anesthesiologist’s thorough evaluation of associated cardiovascular risk. Therefore, it is crucial to closely monitor the patient’s vital signs and ECG upon entering the operating room before anesthetic induction. Various guidelines regarding perioperative management have recommended measuring preoperative BNP and NT-proBNP assays and postoperative daily troponin assays in patients with one or more cardiovascular risk factors. In addition to that, we recommend that perioperative cardiac biomarker assays, such as cTn, CK-MB, BNP, and NT-proBNP, should be performed in patients with risk factors to lower the risk of PMI during surgery.

## Figures and Tables

**Figure 1 medicina-59-01168-f001:**
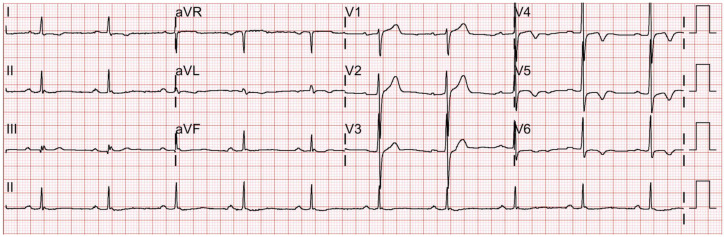
Preoperative 12-lead electrocardiography taken 15 days prior to the scheduled surgery shows normal sinus rhythms and nonapparent precordial T-inversions in V4-6 that were confirmed by a cardiologist.

**Figure 2 medicina-59-01168-f002:**
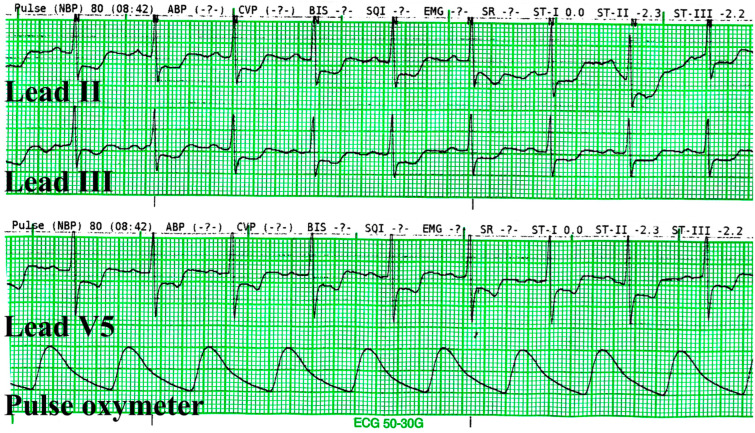
The electrocardiography was captured before sublingual nitroglycerine was administered. The ST-segment depressions were greater than 0.2 mv in leads II, III, and V5.

**Figure 3 medicina-59-01168-f003:**
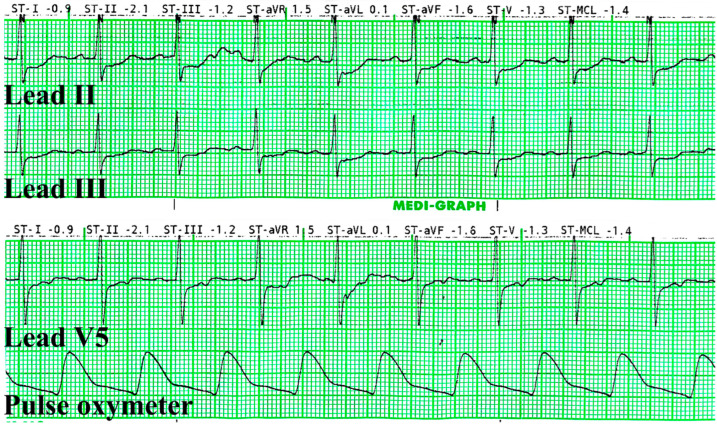
The electrocardiography was captured 10 min after sublingual nitroglycerine was administered. The ST-segment depressions were normalized.

**Figure 4 medicina-59-01168-f004:**
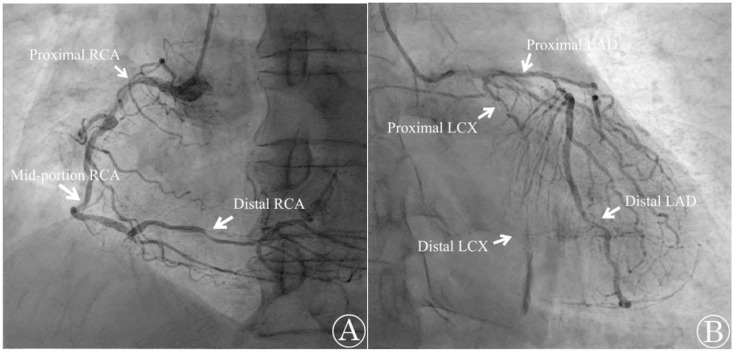
Emergency coronary angiography shows occlusion in multiple sites. (**A**) The right coronary artery (RCA) was found to be occluded by 90% at the proximal segment, 70% at the mid-portion, and 80% at the distal segment; (**B**) the left circumflex artery (LCX) was occluded by 95% at the proximal segment and 100% at the distal segment. The left anterior descending artery (LAD) was occluded 80% at both the proximal and distal portions.

## Data Availability

All information is publicly available and data regarding this particular patient can be obtained upon request from the corresponding senior author.

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
