# Peer review of "Asymptomatic Type 2 Perioperative Myocardial Infarction Detected before Anesthetic Induction in a Patient Undergoing Non-Cardiac Surgery—A Case Report"

_medicina, 2023, doi:10.3390/medicina59061168_

Round 1
Reviewer 1 Report
Ki et al. reported a case of perioperative MI during non-cardiac surgery. The manuscript is well-written but some minor concerns should be addressed:
1- The abbreviations should be defined in their first use. For example, authors should define PMI and NCS in the introduction (the abstract is a separate section and abbreviations in the body text should be defined in first use).
2- Line 49-50: please describe the patient's drug history in more detail (which medications he had taken for DM and HTN).
3- Define abbreviations of Figure 4 in the figure legend.
4- Do you have follow-up information for this patient (more than 3 weeks)? If yes, please add it to the "case" section.
The English language is good enough for publication.
Author Response
Point 1: The abbreviations should be defined in their first use. For example, authors should define PMI and NCS in the introduction (the abstract is a separate section and abbreviations in the body text should be defined in first use).
Response 1: I edited them with expanded words. Perioperative myocardial infarction and non-cardiac surgeries are abbreviated as PMI and NCS, respectively.
Point 2: Line 49-50: please describe the patient's drug history in more detail (which medications he had taken for DM and HTN).
Response 2: A calcium channel blocker, angiotensin II receptor blocker, and hydrochlorothiazide were prescribed by the cardiologist for hypertension. Metformin and combined basal and rapid insulin were prescribed by the endocrinologist for insulin-dependent DM.
Point 3: Define abbreviations of Figure 4 in the figure legend.
Response 3: The right coronary artery, left circumflex artery, and left anterior descending artery are abbreviated as RCA, LCX, and LAD, respectively. Words in figure legends define abbreviations in Figure 4.
Point 4: Do you have follow-up information for this patient (more than 3 weeks)? If yes, please add it to the "case" section.
Response 4: After the patient was discharged from our hospital (3 weeks after CABG), he regularly visits our outpatient cardiothoracic and endocrinology centers. He is well-ambulated with good general conditions. His back pain is being managed through medication and nerve root block procedures. Laboratory findings and electrocardiography were normal. I added the below paragraph in the case description section.
Currently, the patient regularly visits the cardiothoracic surgeon and the endocrinologist for monthly follow-up examinations. During his last two visits, there were no ST changes in 12-lead ECGs, and cardiac biomarkers were within normal ranges, including CK-MB, cTnI, and LDH. Brain natriuretic peptide (BNP) and N-terminal pro-brain natriuretic peptide (NT-proBNP) were not measured. The patient is well-ambulated and his back pain is being managed at our pain clinic center through medication and nerve root block injections. The cardiothoracic surgeon will evaluate the patient’s conditions 12 months after his CABG for elective spine surgery with the neurosurgery department.
Reviewer 2 Report
Dear authors,
That was a nice work! An interesting case report on a clinically relevant topic, not so rarely observed.
I am in favor of its publication since the reader can take home some nice messages! Just some minor comments:
Introduction. Line 27. Please expand NCS upon its first use. You should be consistent with all abbreviations used.
Please provide the medications used by the patient. Was his DM insulin-dependent?
Please provide any other study investigating the Perioperative Ischemic Evaluation issue. It is really crucial to provide specific recommendations on how to treat those patients when the MI is suspected.
Proofreading of the whole manuscript would be recommended.
Author Response
Point 1: Introduction. Line 27. Please expand NCS upon its first use. You should be consistent with all abbreviations used.
Response 1: I edited them with expanded words. Perioperative myocardial infarction and non-cardiac surgeries are abbreviated as PMI and NCS, respectively.
Point 2: Please provide the medications used by the patient. Was his DM insulin-dependent?
Response 2: Yes, his DM is in an insulin-dependent state. As a result, the endocrinologist has prescribed him a metformin and combined basal and rapid insulin. In addition, the cardiologist has prescribed him a CCB, ARB, and diuretics. Here’s an added paragraph:
A calcium channel blocker, angiotensin II receptor blocker, and hydrochlorothiazide were prescribed by the cardiologist for hypertension. Metformin and combined basal and rapid insulin were prescribed by the endocrinologist for insulin-dependent DM.
Point 3: Please provide any other study investigating the Perioperative Ischemic Evaluation issue. It is really crucial to provide specific recommendations on how to treat those patients when the MI is suspected.
Response 3: Defilipis et al and 2022 ESC guideline provides a comprehensive systematic approach to evaluating perioperative myocardial injury. Many articles have addressed the pre or post-operative management of PMI, offering specific recommendations. It consists of coronary interventional or surgical treatment and the use of medications for CAD. However, those recommendations are not easily applicable in intraoperative scenarios. Coronary interventions and CABG are not done simultaneously with ongoing non-cardiac surgery. And using anti-platelet and dual antiplatelet therapy during the intraoperative period should be approached with caution due to the potential risk of increased bleeding. Noteworthy authors such as Ruetzler et al; Landesberg et al; and Alkhatib et al. emphasize the importance of maintaining a patient’s vital sign stable during an operation, which is a key goal for anesthesiologists.
I added and edited the paragraph to clarify the PMI evaluation from pre-, intra-, and post-operative perspectives. The risk factors associated with PMI have been added, in addition to the specific treatment options during the intraoperative period. I have also added a paragraph about the systematic approach to patients with suspected PMI.
Please refer to my edited manuscript.
I’m very pleased and give thanks to you for your reviewing my manuscript.